# Effects of Storage Temperature on Indica-Japonica Hybrid Rice Metabolites, Analyzed Using Liquid Chromatography and Mass Spectrometry

**DOI:** 10.3390/ijms23137421

**Published:** 2022-07-04

**Authors:** Lin Zhu, Yu Tian, Jiangang Ling, Xue Gong, Jing Sun, Litao Tong

**Affiliations:** 1Key Laboratory of Preservation Engineering of Agricultural Products, Ningbo Academy of Agricultural Sciences, Institute of Agricultural Products Processing, Ningbo 315040, China; zhulin0822@163.com (L.Z.); nbnjg@163.com (J.L.); 2Key Laboratory of Agro-Products Processing Ministry of Agriculture, Institute of Food Science and Technology, Chinese Academy of Agricultural Sciences, Beijing 100193, China; tianyu152302@163.com (Y.T.); xuegong1214@163.com (X.G.)

**Keywords:** indica/japonica hybrid rice, liquid chromatography–mass spectrometry, wide-targeted metabolomics, storage temperature, storage performance

## Abstract

The Yongyou series of indica-japonica hybrid rice has excellent production potential and storage performance. However, little is known about the underlying mechanism of its storage resistance. In this study, Yongyou 1540 rice (*Oryza sativa cv. yongyou 1540*) was stored at different temperatures, and the storability was validated though measuring nutritional components and apparent change. In addition, a broad-targeted metabolomic approach coupled with liquid chromatography-mass spectrometry was applied to analyze the metabolite changes. The study found that under high temperature storage conditions (35 °C), Yongyou 1540 was not significantly worse in terms of fatty acid value, whiteness value, and changes in electron microscope profile. A total of 19 key differential metabolites were screened, and lipid metabolites related to palmitoleic acid were found to affect the aging of rice. At the same time, two substances, guanosine 3′,5′-cyclophosphate and pipecolic acid, were beneficial to enhance the resistance of rice under harsh storage conditions, thereby delaying the deterioration of its quality and maintaining its quality. Significant regulation of galactose metabolism, alanine, aspartate and glutamate metabolism, butyrate metabolism, and arginine and proline metabolism pathways were probably responsible for the good storage capacity of Yongyou 1540.

## 1. Introduction

Rice (*Oryza sativa* L.) is one of the major staple foods consumed all over the world, notably in China, India, Indonesia, Bangladesh, Vietnam, Thailand, Myanmar, and the Philippines. These countries contribute most highly to rice production, corresponding to 82% of global rice production and 69% of global rice consumption [1]. At the same time, rice also provides 35–60% of dietary calories for most people in the world [2]. However, in the next few decades, with the increase of population, the demand for food in China and the world will remain a serious challenge. Breeding high-yielding rice varieties and developing high-yielding cultivation techniques are considered two key approaches to address this challenge [3,4]. Indica-japonica hybrid rice is a new type of rice cultivar developed by hybridizing indica as the male parent and japonica as the female parent. Moreover, the hybrid rice plant structure has the ability to support super high yield, and the Yongyou series have a positive effect [5].

The storage of rice is also an important part of its production and transactions, and only with good storage performance can its commercial value be maintained in the commodity transaction link. Starch, protein, lipid, mineral elements and vitamins, are the main nutritional components of rice [6,7,8,9], which is susceptible to microbial contamination and oxidative deterioration during storage, leading to reduced physiological activity, fat oxidation, and protein degradation [10]. The rice quality reduction was usually manifested in the weakening of grain respiration, the reduction of vitality, the change of physicochemical properties and the change of protoplasmic colloid structure [11]. There are many factors affecting the aging of rice, such as the variety, water content, temperature, humidity, storage time, storage conditions, and local climatic conditions.

Yongyou 1540 rice is a new variety jointly cultivated by the research group, and it has been proved that Yongyou 1540 has good storability. However, the underlying mechanisms are still largely unknown, which limits the promotion and application of the products. Metabolomics is the systemic study of the metabolites, that is, of all small molecules in a biological sample, to provide a snap-shot of the ongoing biochemical processes [12,13]. At the same time, it is also the science of qualitative and quantitative analysis of all low molecular weight (<1000) metabolites of biological cells or organisms in a specific physiological period [14].

In this study, in order to reveal the storage mechanism of Yongyou 1540, and to provide theoretical data support for the production and promotion of this series of indica-japonica hybrid rice, Yongyou 1540 was stored at different temperatures. The fatty acid and whiteness values of each sample were determined, and the rice surface was observed with an electron microscope. In addition, the changes of Yongyou 1540 metabolites under different storage conditions were investigated using a broad-targeted metabolomic technology based on LC-MS/MS. After multivariate statistical analysis such as principal component analysis, cluster analysis, and path analysis, significant differential metabolites were screened out. Finally, through enrichment analysis of metabolic pathways, the most relevant metabolic pathways were found. The results showed that the key differential metabolites of Yongyou 1540 that were significantly up-regulated were guanosine 3′,5′-cyclic phosphate, pipecolic acid and GABA. Based on the properties and effects of these substances, it is speculated that these substances increase the resistance of rice to relatively harsh storage conditions, thereby delaying the aging process and maintaining its quality.

## 2. Result

### 2.1. Measurement Results of Fatty Acid Value and Whiteness Value of Rice

As shown in Figure 1A, the higher the storage temperature, the higher was the fatty acid value (from 0.085 to 0.350 g kg^−1^). According to the Chinese national standard GB/T20569-2006, the fatty acid value of fresh rice should be less than 0.250 g kg^−1^ and that of stored rice should be less than 0.350 g kg^−1^ [15]. Even under the storage condition of 35 °C, the fatty acid value met the requirements, and the degree of quality deterioration of the Yongyou rice was not significant.

The test results of whiteness value are shown in Figure 1B. The whiteness value represents the level of whiteness of the rice surface. Obviously, with the prolongation of storage time, the whiteness value of rice at different storage temperatures showed a significant downward trend, and then entered a stable stage and continued to decrease slowly. Due to the influence of storage temperature, the higher the temperature, the greater is the decrease of the whiteness value. However, the whiteness values (from 68% to 76%) are still in the reasonable whiteness value range [16].

### 2.2. Electron Microscope Section Observation Results

The section images of rice at different storage temperatures were observed under different magnifications (left: 1 mm, right: 200 μm) (Figure 2). Through observation, it can be seen that with the continuous increase of the ambient temperature of the rice, the width and depth of the cracks on the cut surface of the rice samples increased significantly by the 90th day of storage. This may be due to the influence of temperature in the early storage period. The loss of nutrients was accelerated, as well as the rate of surface water loss and the degree of deterioration. On the 180th day of storage, there was no obvious further deterioration compared with the previous period. This may be due to the fact that when kept under unsuitable storage conditions for a long time, a certain metabolic system was activated by the rice itself, which enhanced its resistance to stress and storage performance.

### 2.3. Metabolite Characterization of Samples

The LC-MS-based wide-targeted metabolomics approach was employed in the study. By matching the substance database with conditions such as retention time and mass-to-charge ratio, the substance could be qualitatively identified [17]. A total of 177 metabolites were identified, as shown in Appendix A. In terms of quantity, alkaloids account for 14.12%, amino acids and their derivatives account for 9.60%, phenols and their derivatives account for 8.74%, organic acids and their derivatives account for 8.47%, fatty acyl groups account for 6.7%, and flavonoids account for 7.3%, carbohydrates and their derivatives accounted for 6.21%, terpenoids accounted for 6.21%, and other small amounts of substances such as steroids and their derivatives, plant hormones, coumarin, purines, pyridines, etc. accounted for 12.99%. Alkaloids accounted for the largest proportion, followed by amino acids and their derivatives, while steroids and their derivatives and terpenes were also present.

### 2.4. Principal Component Analysis

The PCA score scatter plot of all samples (including QC samples) is shown in Figure 3. The value of R^2^X in the scatter plot of the PCA model was 0.527, which further indicated that the experimental model was not over-fitting, and the experimental results were reliable. It could be clearly observed that there was a significant difference between HT_6, LT_6, NT_6 groups and the CK group, while the difference between the HT_6 group and CK group was the most significant. The comparisons between the LT_6 group and the CK group were close in degree of difference, suggesting that storage temperature was an important factor affecting the metabolic activity of rice. However, there are a certain number of samples with no obvious changes in the principal component differences. This may indicate that the rice of this variety produces a certain substance that is self-regulating even under inappropriate storage conditions.

### 2.5. OPLS-DA Analysis

The abscissa t[1]P in the OPLS-DA scored graph represented the predicted principal component score of the first principal component, showing the difference between sample groups, and the ordinate t[1]O represented the orthogonal principal component score, showing the difference within the sample group. In the present study, OPLS-DA was modeled for the classification between the HT_6 and CK_6 groups, and the HT_6 and LT_6 groups, respectively. From the results of the score plot, it can be seen that the HT_6 group was very significantly distinguished from the CK group and the LT_6 group, and the samples were all within the 95% confidence interval (Hotelling’s T-squared ellipse), indicating that the storage temperature changed the metabolites of the rice samples. According to the statistics, HT_6 vs. CK groups Q^2^ = 0.701, HT_6 group vs. LT_6 group Q^2^ = 0.654, indicating that if new samples are added to the model, a relatively approximate distribution will be obtained. The permutation test randomly changed the arrangement order of the categorical variable Y, and established the corresponding OPLS-DA, modeled multiple times (*n* = 200 times) to obtain the R^2^Y and Q^2^ values of the random model. The intercept of the regression line of Q^2^ and the vertical axis was less than zero. At the same time, with the gradual decrease of the permutation retention, the proportion of the permuted Y variable increased, and the Q^2^ of the random model gradually decreased. The results above showed that the original models of HT_6 vs. CK groups and HT_6 vs. LT_6 groups could meet the acceptable requirements.

The Q^2^ values of NT_6 vs. CK groups, NT_6 vs. LT_6 groups, HT_6 vs. NT_6 groups were close to 0.5, indicating that there were differences between sample groups, but the degree of difference was not significant. The Q^2^ < 0.5 of the LT_6 group and the CK group indicated that the metabolic difference between the two samples was not significant within a certain range. This phenomenon shows that the metabolic activity of Yongyou 1540 rice is slow under the storage condition of 15 °C (see Figure 4).

### 2.6. Screening and Analysis of Differential Metabolites with Significant Changes in Content

In order to explore the effects of different storage temperatures on rice metabolites and quality, the VIP value of the OPLS-DA model (threshold > 1) and the *p*-value of Student’s *t*-test (threshold < 0.05) were used to screen for differential metabolites. The results showed that there were 105 differential metabolites in HT_6 vs. CK group, 99 differential metabolites in the HT_6 vs. LT_6 group, 81 differential metabolites in HT_6 vs. NT_6 group, 93 differential metabolites in NT_6 vs. CK group, 87 differential metabolites in NT_6 vs. LT_6 group and 49 differential metabolites in LT_6 vs. CK group. The differential metabolites obtained through the above analysis often have the results and functional similarity/complementarity in biology, or are positively regulated/negatively regulated by the same metabolic pathway, showing similar or opposite expression characteristics between different experimental groups. Hierarchical cluster analysis was used to help explore similarities and differences between different groups. Figure 5 illustrated the thermodynamic diagram of the hierarchical cluster analysis of experimental groups at different storage temperatures [18]. The differential metabolites of the above experimental groups were also subjected to matchstick analysis and chord analysis. Matchstick analysis and chord analysis help to further screen the most closely related differential metabolites. The identification of differential metabolites can help to find the relevant metabolic pathways. HT_6 vs. CK groups were used as the core of the analysis. The research groups compared the results of their matchstick analysis and chord analysis, and screened for 19 key differential metabolites. These 19 substances are shown in Table 1. Among the above 19 differential metabolites, 8 substances were distinct differential metabolites in the HT_6 vs. NT_6 groups, and 14 substances were distinct differential metabolites in the HT_6 vs. LT_6 groups. The metabolites of LT_6 and CK were so similar that it was difficult to find the differential metabolites.

This result indicates that some reactions are only excited at the higher temperature conditions where HT_6 and NT_6 are located. The two groups had 8 identical differential metabolites, indicating that when the storage temperature was higher, the corresponding metabolic pathways would be activated to produce corresponding metabolites. The only other 11 differential metabolites expressed in the HT_6 vs. CK groups were those produced by the corresponding stimulated metabolic pathways in this rice variety under relatively harsher storage conditions. Among these 11 differential metabolites, adenosine 2′,3′−cyclic phosphate, guanosine 3′,5′−cyclic monophosphate, (S) −2−aceto−2−hydroxybutanoate, streptozotocin, palmitoylethanolamide, pimelic acid, pipecolic acid, these 7 substances are up-regulated. Stachyose, maltotetraose, kojibiose, and denudatine were down-regulated. The first three substances that were down-regulated are carbohydrates, and the down-regulation of carbohydrates indicated that the metabolism of energy substances was significant under high temperature storage conditions. HT_6 vs. LT_6 groups had the highest number of overlapping differential metabolites with HT_6 vs. CK groups. The HT_6 vs. CK group was the focus of the analysis, with a larger temperature span and wider coverage.

### 2.7. Effects of Storage on Metabolic Pathways of Rice

The research group mapped 19 key differential metabolites screened by the HT_6 vs. CK groups to authoritative metabolite databases such as KEGG and PubChem, and found all the pathways involved in the regulation of these differential metabolites (Figure 6 and Figure 7). Through a comprehensive analysis of metabolic pathways (including enrichment analysis and topology analysis), the key pathways most associated with metabolite differences were found. As shown in Figure 6A, five of the most relevant pathways were located, including valine, leucine and isoleucine biosynthesis, galactose metabolism, lysine biosynthesis and degradation, and glutathione metabolism. Meanwhile, the differential metabolites with the highest correlation between the HT_6 and LT_6 groups were also located, and the most closely related metabolic pathways were identified, as shown in Figure 6B. It can be seen that the difference in metabolic pathways between the two experimental groups is only in the high expression of starch and sucrose metabolism in the HT_6 and LT_6 groups. The metabolic activity of rice stored at 15 °C was higher than that of rice stored at −80 °C. Energy metabolism is the normal physiological activity of rice.

## 3. Discussion

The basic quality of You 1540 was evaluated from three aspects: fatty acid value, whiteness value, and electron microscope section observation. According to the experimental results, the following conclusions can be drawn: At different storage temperatures, the changing trend of fatty acid value is different. The higher the storage temperature, the more obvious was the increase of fatty acid value. However, under the storage condition of 35 °C, the fatty acid value of rice was also within 0.350 g kg^−1^, indicating that the rice was slightly unsuitable for storage at this temperature. However, the degree of quality deterioration of Yongyou rice was not significant. Similarly, for the whiteness value of Yongyou 1540, the research team believes that the change of whiteness value was within an acceptable range after evaluation, and there was no obvious deterioration. On the other hand, the section of Yongyou 1540 was observed under an electron microscope with a magnification of 200 μm. On the 90th day, the depth and width of surface cracks of rice stored at 35 °C were indeed more obvious than those of rice at the other two storage temperatures. However, on the 180th day, the change of rice cracks was not obvious under different storage temperatures, and the degree was lower than that on the 90th day. Based on the above conclusions, it can be speculated that Yongyou 1540 performed certain metabolic activities and regulated certain pathways under unsuitable storage conditions, and this activity was related to its storage resistance and stress resistance.

To further analyze and explore the effects of metabolites on the storage performance of Yongyou 1540, the research group used liquid chromatography and mass spectrometry-based broad-target metabolomics techniques to isolate a total of 719 metabolites. Of these, 615 were recorded in public databases. Principal component analysis (PCA) and OPLS-DA analysis were used in this study. PCA analysis is a statistical method that transforms a set of observed, possibly correlated variables into linearly uncorrelated variables (i.e., principal components) through an orthogonal transformation. The PCA method can reveal the internal structure of the data and thus better explained the data variables between different groups [19]. In fact, LC-MS-based metabolomics data are characterized by high dimensionality (many types of metabolites detected) and small sample size (too few samples detected). These variables contain not only differential variables related to categorical variables, but also a large number of interrelated indifference variables [20]. Therefore, the OPLS-DA method, a supervised pattern recognition technique that outperforms PCA in class discrimination, was used to find distinct metabolites between each pair of groups. Through OPLS-DA analysis, orthogonal variables unrelated to categorical variables in metabolites were filtered out, and non-orthogonal variables and orthogonal variables were analyzed separately, so as to obtain more reliable information on the difference and influence degree between metabolite groups. PCA analysis can provide support in the acquisition of related metabolites, and OPLS-DA analysis can support the screening and identification of key differential metabolites. At the same time, the 19 most important differential metabolites were finally selected by using matchstick analysis and chord analysis to assist the screening.

Different storage temperatures were set in the experiments. The maximum storage temperature of 35 °C was used as the core of the exploration, and the key differential metabolites were analyzed. A certain number of differential metabolites overlapped with differential metabolites screened at other storage temperatures. In order to explore the metabolic activities and reactions of Yongyou 1540 at higher storage temperatures, the research team chose to analyze 11 non-overlapping differential metabolites one by one. Palmitoylethanolamide (PEA) is a natural amide of ethanolamine and palmitic acid. Pimelic acid exists in the free acid form and is synthetically assembled from fatty acids. These two substances are metabolites produced by the aging phenomenon caused by lipid metabolism in rice during storage. Guanosine 3′,5′−cyclic monophosphate (cGMP), a second messenger discovered in the 1960s, is found in both prokaryotes and eukaryotes [21]. The molecule cGMP is synthesized from guanosine triphosphate (GTP) by guanylate cyclase enzymes (GCs) and is involved in various cellular responses, such as protein kinase activity, cyclic nucleotide gated ion channels and cGMP regulated cyclic nucleotide phosphodiesterases [22,23]. There is a positive correlation between the accumulation of cGMP in plants and various developmental processes and responses to abiotic and pathogenic stresses. Multiple groups have demonstrated that both NO-dependent and NO-independent cGMP signaling pathways are important in the activation of defense responses during biological stress [24,25,26,27,28]. Furthermore, NO-cGMP-dependent signaling pathways have been reported to be involved in adventitious root development [29,30], stomatal closure during abiotic and biotic stresses [31,32], protein phosphorylation [33,34] and transcriptional regulation [35]. Internal cGMP production in rice involves various signaling processes in the plant, especially in control of stomatal pore size. In a high-temperature storage environment, this is important for surviving water shortages, and this key metabolite helps rice maintain water content in the body and delay quality deterioration. Pipecolic acid is L-Pip (hereinafter referred to as Pip); the widespread occurrence of the non-protein amino acid L-Pip in plants, animals, fungi, and microorganisms and its biosynthetic origin from Lys in plants and animals was realized in the 1950s [36,37]. Pip, a common lysine catabolite in plants and animals, is a key regulator of induced plant immunity. Pip is one of several key metabolic mediators for the induction of resistance [38,39,40]. Based on the above analysis of differential metabolites with significant effects, it can be speculated that under the storage condition of higher temperature, Yongyou 1540 significantly accumulated pipecolic acid, which induced rice to improve its own stress resistance to relatively harsh conditions and maintain its own quality.

In conclusion, a broad target metabolomics technique based on liquid chromatography and mass spectrometry was used in this study. The changes of metabolites of Yongyou 1540 at different storage temperatures were fully analyzed. Through the screening analysis and metabolic pathway mapping of key differential metabolites, Yongyou 1540 was shown to produce palmitoleic acid-related lipid metabolites that are associated with rice aging when stored at 35 °C. At the same time, the key differential metabolites of Yongyou 1540 were significantly up-regulated: guanosine 3′,5′-cyclic phosphate, pipecolic acid and GABA.

Based on the properties and effects of these substances, it is speculated that these substances increase the resistance of rice to relatively harsh storage conditions, thereby delaying the aging process and maintaining its own quality [41,42]. The results show that Yongyou 1540 has self-regulating ability under unsuitable storage conditions, and has good storage performance, which is beneficial to maintain the quality of production and transaction links. Under the increasingly heavy pressure for global food staples, this research can provide a theoretical and scientific basis for the popularization and production of Yongyou 1540.

## 4. Materials and Methods

### 4.1. Rice Materials and Sampling

Yongyou 1540 rice was harvested in Xiangshan County, Ningbo City, Zhejiang Province. The harvested rice was dried (RH 13–14%), and then ground into first-class rice according to GB/T1354 standard. All samples were divided into four groups. The control (CK) was frozen at −80 °C immediately after grinding. After the other three groups were milled, the rice was put into PE bags (2.5 kg per bag) and stored in a constant temperature and humidity box at 15 °C (LT_6), 25 °C (NT_6), or 35 °C (HT_6) (RH 60%), which were used as the experimental groups. There were six samples in parallel for each group. All samples were sent for testing after being stored for 6 months.

### 4.2. Materials and Reagents

Methanol (CAS: 67-56-1) and acetonitrile (CAS: 75-05-8) were supplied by CNW Technologies, and SIGMA brand formic acid reagent (CAS: 64-18-6) was used. The purity of the above reagents was all LC-MS grade. Ultrapure water was prepared in-house by a Milli-Q Integral water purification system (Millipore, Bedford, MA, USA). During the experiment, Sciex’s model ExionLC AD ultra-high performance liquid phase and QTrap 6500+ high-sensitivity mass spectrometer were used. In addition, a Thermo centrifuge (Heraeus Fresco17), a Merck Millipore water purifier (Clear D24 UV) and a Waters chromatographic column (ACQUITY UPLC HSS T3 1.8 μm 2.1 ∗ 100 mm) were also used in the experiments.

### 4.3. Metabolites Extraction

The freeze-dried samples were crushed with a mixer mill for 240 s at 60 Hz. 400 mg aliquots of individual samples were precision weighed and then transferred to an Eppendorf tube. After addition of 2000 μL of extract solution (methanol/water = 3:1, precooled at −40 °C, containing internal standard), the samples were vortexed for 30s, homogenized at 40 Hz for 4 min, and sonicated for 5 min in an ice-water bath. After repeated homogenization and sonication for 3 times, the samples were extracted over night at 4 °C on a shaker. The resultant mixtures were centrifuged at 12,000× *g* rpm (RCF = 13,800× *g*, R = 8.6 cm) for 15 min at 4 °C, and the supernatant was carefully filtered through a 0.22 μm microporous membrane, and transferred to 2 mL glass vials. Moreover, 20 μL aliquots of each sample were pooled and prepared in parallel with other rice samples to yield quality control (QC) samples. All samples were stored at −80 °C until the UHPLC-MS analysis.

### 4.4. LC-MS Analysis

The UHPLC separation was carried out using an EXIONLC System (Sciex). The mobile phase A was 0.1% formic acid in water, and the mobile phase B was acetonitrile. The column temperature was set at 40 °C. The auto-sampler temperature was set at 4 °C and the injection volume was 2 μL. A Sciex QTrap 6500+ (Sciex Technologies, Framingham, MA, USA), was applied for assay development. Typical ion source parameters were: IonSpray Voltage: +5500/−4500 V, Curtain Gas: 35 psi, Temperature: 400 °C, Ion Source Gas 1:60 psi, Ion Source Gas 2: 60 psi, DP: ±100 V. In fact, due to the large sample size of this study, the detection task lasts for a long time, so it is very important to monitor the stability of the instrument and whether the signal is normal during the detection process in real time. To ensure the accuracy of the experimental results, the original data included 3 quality control (QC) samples and 24 experimental samples. QC samples were made by mixing 20 μL of each experimental sample. The retention times and peak areas of the QC samples TIC overlap well, indicating good instrument stability. In addition, the retention time and peak area of the internal standard 2-chlorophenylalanine response were very stable, indicating that the instrument’s data acquisition stability was very good (Appendix A). The experimental equipment is reliable and the data is credible [43,44,45,46,47,48,49,50,51].

### 4.5. Data Analysis

SCIEX Analyst Work Station Software (Version 1.6.3, Framingham, MA, USA) was employed for MRM data acquisition and processing. MS raw data (.wiff) files were converted to the TXT format using MSconventer. In-house R program and database were applied to peak detection and annotation. After obtaining the collated data, the SIMCA software (V16.0.2, Sartorius Stedim Data Analytics AB, Umea, Sweden) was used to screen for differential metabolites using multivariate statistical analysis, such as principal component analysis (PCA), orthogonal partial least squares discriminant analysis (OPLS-DA), Student’s *t*-test, and variable importance in projection (VIP) principal components of the OPLS-DA model. The card value standard used in this project is that the *p*-value of the Student’s *t*-test is less than 0.05, and the VIP of the first principal component of the OPLS-DA model greater than 1. According to the expression profiles of differential metabolites, the changes of metabolites between groups can be summarized, and the useful information behind them can be mined in combination with the disciplinary background. For example, if some metabolites have the same or different variation trends among groups, combining metabolic pathways can help to mine important metabolic pathways and regulatory relationships. Differential metabolites were annotated in CAS and KEGG databases based on retention time and mass-to-charge ratio (*m*/*z*). Afterwards, through comprehensive analysis of differential metabolite pathways (including enrichment analysis and topology analysis), the pathways were further screened to identify the key pathways of the most relevant differential metabolites [52,53,54].

### 4.6. Determination of Rice Quality

#### 4.6.1. Fatty Acid Value (Petroleum Ether Extraction Method)

Nutrients such as starch, protein, moisture, and lipids in rice itself will deteriorate due to the influence of the environment and internal factors during the storage process. Fatty acid value is an important indicator to measure the quality of rice. According to the Chinese national standard GB/T5510-2011 “Grain and Oil Inspection—Determination of Fatty Acid Value of Grain and Oilseeds”, the research team measured the fatty acid value of rice to judge the degree of deterioration of rice at different storage temperatures. Rice was crushed by grinder under different storage conditions (the groups needed cleaning and were passed through a 1.0 mm round hole sieve). A sample of 10 g (precision 0.01 g) was weighed and placed in a 250 mL conical flask. 50 mL petroleum ether was added to the pipette, and the plug was added before shaking for several seconds. The plug was opened and deflated, and then the bottle was closed and the oscillator was shaken for 10 min. Next steps: Take off the conical bottle, tilt and stand for 1–2 min, put filter paper into the short neck glass funnel to filter. Go to the first few drops of filtrate, use colorimetric tube to collect filtrate more than 25 mL, cover and save. (Tight timing is important: placed at 4 °C, use within 24 h). 25 mL filtrate was removed in 150 mL conical flask with a pipette, 75 mL 50% ethanol solution was added into the measuring cylinder, 4–5 drops of phenolphthalein indicator was added, shaken, titrated with potassium hydroxide solution until the lower ethanol solution was slightly red, 30 s did not fade, the titration solution volume (*V*_1_) was recorded, and 25 mL petroleum ether was used as the blank control group, the titration solution volume (*V*_0_) was recorded.

The acid value (*A_K_*) formula is as follows:$$A_{K}=\left(V_{1}-V_{0}\right)*c*56.1*\frac{50}{25}*\frac{100}{m\left(100-w\right)}*100$$

*c*-potassium hydroxide concentration (mol/L)

*m*—Sample mass (g)

*w*—Sample moisture mass (per 100 g)

56.1—Potassium hydroxide molar mass (g/mol)

50—The volume of the extraction solution used to extract the sample(mL)

25—Volume of sample extract for titration (mL)

100—Converted to the mass of 100 g dry sample (g)

#### 4.6.2. Whiteness Value

The whiteness of rice under different storage conditions was measured with a whiteness meter, and each sample was measured three times.

#### 4.6.3. Observation of Appearance Features in Slices under Electron Microscope

Section observation (2 mm thick cross section), sampling at 90 d and 180 d, respectively. Instrument name of the microscope used in this project: Thermal Field Emission Scanning Electron Microscope (Zeiss G300); Model: GeminiSEM 300.

## Figures and Tables

**Figure 1 ijms-23-07421-f001:**
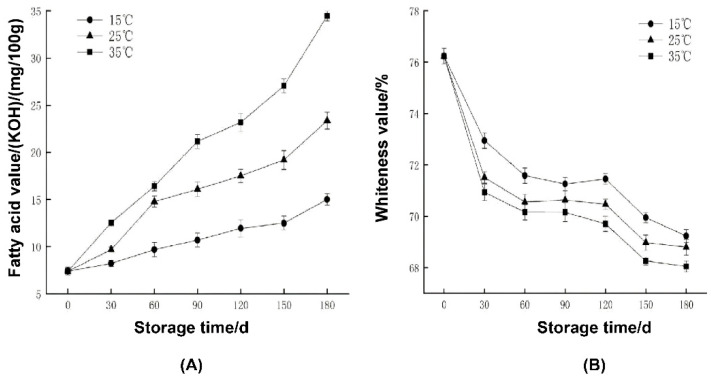
Changes in fatty acid value (**A**) and whiteness value (**B**) at different storage temperatures.

**Figure 2 ijms-23-07421-f002:**
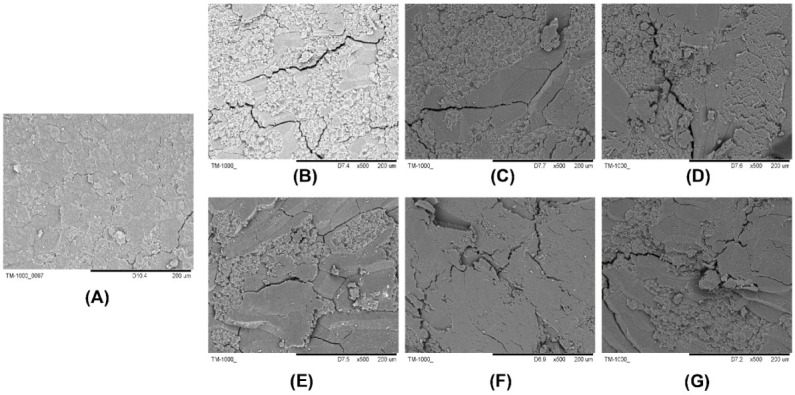
Electron microscope profiles of rice (200 μm) at different storage temperatures. (**A**) Morphological images of rice stored on day 0. (**B**–**D**) are the morphological images of rice on the 90th day of storage at 15 °C, 25 °C, and 35 °C storage temperature, respectively. (**E**–**G**) are the morphological images of rice on the 180th day of storage at 15 °C, 25 °C, and 35 °C storage temperature, respectively.

**Figure 3 ijms-23-07421-f003:**
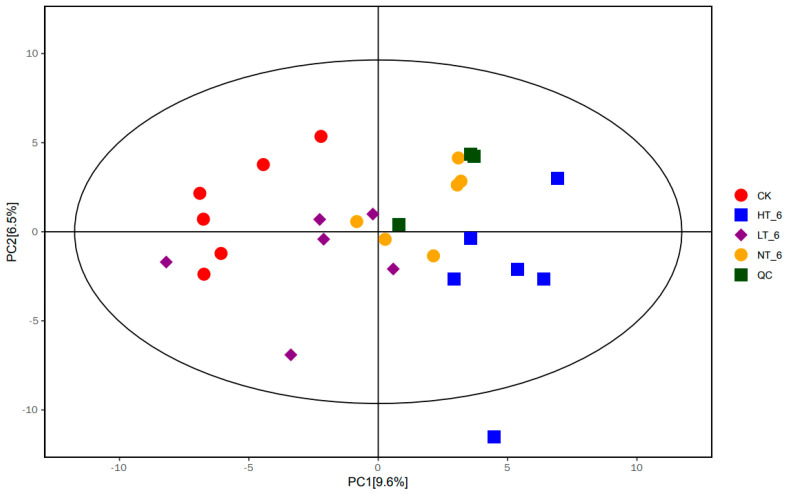
Scatter plot from PCA model for CK_6, LT_6, NT_6, HT_6, and QC groups. The abscissa PC[1] and ordinate PC[2] in the figure represent the scores of the first and second principal components, respectively. Each scatter represents a sample, and the color and shape of the scatter represent different groups. The samples were basically within the 95% confidence interval, and the QC group was tightly gathered and close to the middle of all samples, suggesting great system stability within the entire measurement queue. R^2^X represents the model’s interpretation of the X variable and Q^2^ represents the predictability of the model. The closer the two metrics are to 1, the better the model performs and the higher the interpretability.

**Figure 4 ijms-23-07421-f004:**
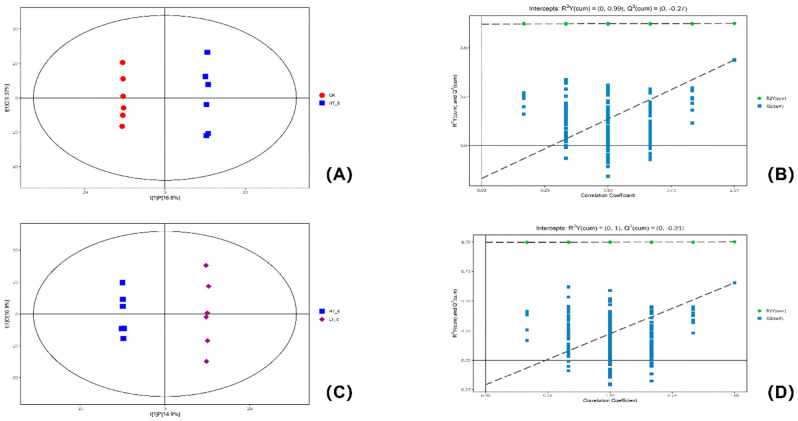
The scatter plot and permutation tested results of the OPLS-DA model for the HT_6 vs. CK groups, and the HT_6 vs. LT_6 groups. (**A**) The scatter plot of the OPLS-DA modeled on the HT_6 vs. CK groups; (**B**) The permutation test of the OPLS-DA modeled on the HT_6 vs. CK groups; (**C**) The scatter plot of the OPLS-DA modeled on the HT_6 vs. LT_6 groups; (**D**) The permutation test of the OPLS-DA modeled on the HT_6 vs. LT_6 groups.

**Figure 5 ijms-23-07421-f005:**
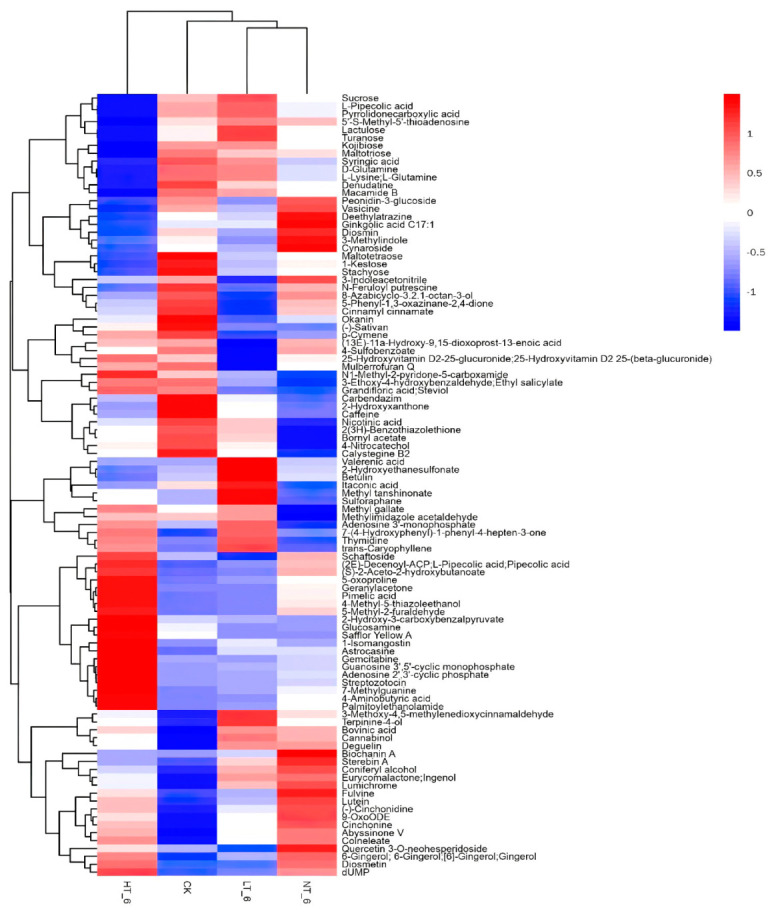
Hierarchical cluster analysis heatmap for all the experimental groups. The abscissa in the figure represents different experimental groups, the ordinate represents the differential metabolites compared in the group, the color blocks of different positions represent the relative expression levels of the metabolites at the corresponding positions, with the red indicating high expression of the substance, and the blue indicating low expression of the substance.

**Figure 6 ijms-23-07421-f006:**
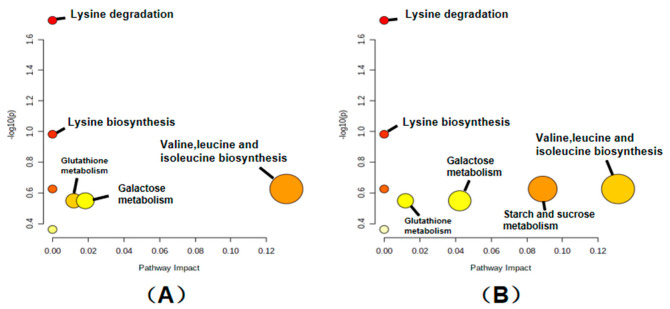
The key differential metabolites of the HT_6 group were mapped to metabolic pathways to analysis, and the results of the metabolic pathway analysis are shown in the bubble chart. (**A**) represents HT_6 vs. ck groups, (**B**) represents HT_6 vs. LT_6 groups. Each bubble in the bubble diagram represented a metabolic pathway. The abscissa of the bubble and the size of the bubble represent the size of the pathway influencing factor in the topology analysis. The larger the scale, the greater the influencing factors. The ordinate where the bubble is located and the color of the bubble represented the *p* value of the enrichment analysis (taking the negative natural logarithm, −ln(p), the darker the color, the smaller the *p* value, and the more significant the enrichment).

**Figure 7 ijms-23-07421-f007:**
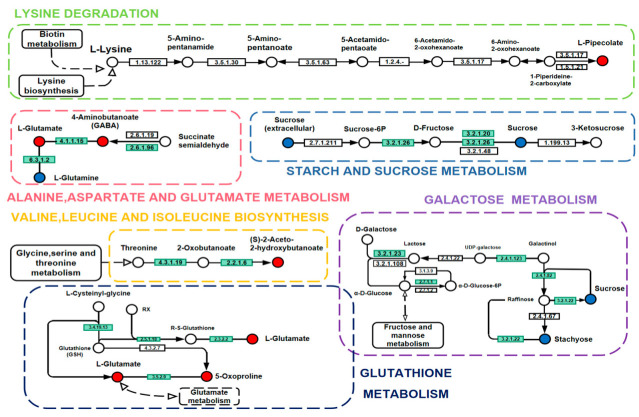
The key differential metabolites of HT_6 vs. CK groups and HT_6 vs. LT_6 groups were mapped to metabolic pathways, and the results of metabolic pathway analysis are shown in the KEGG pathway results.

**Table 1 ijms-23-07421-t001:** Key differential metabolites in HT_6 group.

No.	Compound Name	Rt(min)	KEGG_ID	EXACT_MASS	VIP	*p*-Value	Fold Change	LOG_ Foldchange
**1**	Adenosine 2′,3′−cyclic phosphate	1.86	C02353	329.0525	2.192	0.0001337	30.64	4.937
**2**	(S)−2−Aceto−2−hydroxybutanoate	0.66	C06006	146.0579	2.258	0.00002167	3.301	1.723
**3**	Pimelic acid	0.55	C02656	160.0736	2.131	0.00008361	3.421	1.774
**4**	Pipecolic acid	1.47	C00408	129.0790	2.303	0.0002440	3.809	1.930
**5**	9−OxoODE	11.98	C14766	294.2195	1.504	0.0005610	3.886	1.958
**6**	Palmitoylethanolamide	12.61	C16512	299.2824	2.259	0.0008559	4.063	2.023
**7**	4−Methyl−5−thiazoleethanol	2.66	C04294	143.0405	2.419	0.000003395	5.152	2.365
**8**	5−Methyl−2−furaldehyde	0.76	C11115	110.0368	2.329	0.00000001875	5.440	2.444
**9**	5−Oxoproline	0.56	C01879	129.0426	2.238	0.0002517	6.411	2.681
**10**	7−Methylguanine	1.56	C02242	165.0651	2.425	0.000007288	10.393	3.378
**11**	Streptozotocin	0.72	C07313	265.0910	2.025	0.0002545	16.726	4.064
**12**	Guanosine 3′,5′−cyclic monophosphate	2.26	C00942	345.0474	2.135	0.00002303	18.115	4.179
**13**	4−Aminobutyric acid	0.66	C00334	103.0633	2.062	0.000000003593	21.814	4.447
**14**	d−Glutamine	0.66	C00819	146.0691	2.373	0.0005950	0.2028	−2.302
**15**	l−Glutamine	0.64	C00047; C00064	146.1055	2.319	0.0005209	0.2270	−2.139
**16**	Denudatine	1.59	C08680	343.2511	2.327	0.000001120	0.3601	−1.474
**17**	Maltotetraose	0.64	C02052	666.2219	1.865	0.0026925	0.3738	−1.420
**18**	Stachyose	0.73	C01613	666.2219	1.865	0.0026925	0.3738	−1.420
**19**	Kojibiose	0.72	C19632	342.1162	2.316	0.0002408	0.4181	−1.258

## Data Availability

Not applicable.

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
