# Peer review of "Effects of Storage Temperature on Indica-Japonica Hybrid Rice Metabolites, Analyzed Using Liquid Chromatography and Mass Spectrometry"

_ijms, 2022, doi:10.3390/ijms23137421_

Round 1
Reviewer 1 Report
The aim of this study is to evaluate metabolic changes in in indica-japonica hybrid rice during storage at various temperature. Grain was stored at 15, 25 and 35oC, while control was frozen in liquid nitrogen.
In my opinion, manuscript is badly planned and prepared, I have these suggestions:
Control sample can’t be kept frozen, while other samples are kept in storages with different temperature. In this case you don’t have dynamics in control sample, just starting point. Grain for control should have been kept at temperature used in conventional storages, in this manuscript I suggest using sample at 15 oC as control.
Authors must explain necessity to use 1 mm and 0,2 mm magnification for observation of grain damage together, maybe one of them is enough, and which one?
Figures must have units at axis.
Table 1 should be moved to supplementary material.
Figures in supplementary material should have names and explanation.
Discussion is very narrow; authors mainly repeat result section. Discussion must be expanded and obtained results compared to other authors data.
I am not native English speaker; however English language is bad even for me.
Plants must have Latin name when mentioned first time in text.
In general, manuscript could be published after major revision.
Author Response
Response to Reviewer 1 Comments
Point 1: The aim of this study is to evaluate metabolic changes in in indica-japonica hybrid rice during storage at various temperature. Grain was stored at 15, 25 and 35oC, while control was frozen in liquid nitrogen. In my opinion, manuscript is badly planned and prepared, I have these suggestions:
Control sample can’t be kept frozen, while other samples are kept in storages with different temperature. In this case you don’t have dynamics in control sample, just starting point. Grain for control should have been kept at temperature used in conventional storages, in this manuscript I suggest using sample at 15 oC as control.
Response 1: Thanks for your suggestion. In the current study, four groups including rice stored at 15, 25 ,35 oC, and -80 oC were designed, our original intention was to find the distinct differential metabolites between 35oC and -80 oC groups, and then to the find the change rule of different metabolites at different temperatures. Your suggestion gives us a better idea, the OPLD-DA method were employed to explore the differential metabolites between 35oC and 15 oC groups and we found 14 common differential metabolites. The results further elucidated the mechanism of rice storage tolerance, and the corresponding changes have been made in the main text.
Point 2: Authors must explain necessity to use 1 mm and 0.2 mm magnification for observation of grain damage together, maybe one of them is enough, and which one?
Response 2: Thanks for your suggestion. Compared to 1 mm magnification, 0.2 mm magnification was enough to obseave the changes of rice quality under different storage time and temperatures. In Figure 2, pictures with magnification of 0.2 mm ere retained, and the corrsponding pictures with 1 mm magnification were removed.
Point 3: Figures must have units at axis.
Response 3: Thanks for your suggestion. In Figure 1 and Figure 5, the units of the coordinate axis were supplied.
Point 4: Table 1 should be moved to supplementary material.
Response 4: Done. Please see Table S1 in supplementary material.
Point 5: Figures in supplementary material should have names and explanation.
Response 5: Done. We summarized the figures of supplementary material in a PDF file, and the names and explanation of each figure were added.
Point 6: Discussion is very narrow; authors mainly repeat result section. Discussion must be expanded and obtained results compared to other authors data.
Response 6: We agree with the reviewer. Detailed description and some discussions have been added in the main text. Please see page 12, lines 24-47; page 13, lines 19-22.
Point 7: I am not native English speaker; however English language is bad even for me.
Response 7: We have checked the language throughout our manuscript carefully, and my foreign friend helped us polish this article. We hope the revised version will meet the standard of International Journal of Molecular Sciences.
Point 8: Plants must have Latin name when mentioned first time in text.
Response 8: Thanks for your suggestion.The Latin name of the rice material used in this paper is oryza sativa cv. yongyou 1540, which were added in the main text. Please see page 1,lines 3.
Reviewer 2 Report
In the present work Zhu et al. have analyzed the effects of storage temperature on indica-japonica hybrid rice metabolites, using liquid chromatography and mass spectrometry. The work is technical sound and the authors utilized appropriate techniques of analysis. The results are supported by the data and supply useful conclusion. There are some typewriting errors and some sentences are rambling. The essential problems of this work is lack of novelty and originality.
Some others criticisms are:
- In the abstract is not linear. I don’t understand the meaning of the sentence: “ Studies have shown that using excellent 1540 will produce substances that adjust self-resistance properties under unsuitable storage conditions to maintain its own quality and have good storage capacity.” please clarify. If are results of other studies, the authors have to add references, but I don’t think it is suitable for an abstract.
- The authors have performed a lot of different analysis, based on national and international protocols to analyzed the quality of the rice, along with Electron microscope profiles of rice, why the authors don’t mention them in abstract and focus only on liquid chromatography and mass spectrometry? This last technique are excellent but they aren’t the only one available and resolutive for the determination of the quality of the products. Please update the abstract and revise all the work accordingly.
-Please increase the quality of figure 6. It not possible to read the write present in the graph.
- Are the authors sure that the butanoate metabolism are characteristic of the rice metabolism and not the results of microorganism-yeast metabolism of the rice and there is a contamination by these microorganisms? The change in amino acids and the increase of pipecolic acid (Pip). Moreover the authors declare: “The occurrence of Pip is common in the plant kingdom, and pathogen-induced accumulation of Pip has been reported to occur in rice…..). An analysis of microorganism contamination must be performed.
-The first part of the results shows a style different from the rest of the manuscript, please change.
-In my opinion the conclusion section must be reported in a separate section and substantially improved to better explain the obtained results and their potentiality.
Author Response
Response to Reviewer 2 Comments
Point 1: In the abstract is not linear. I don’t understand the meaning of the sentence: “ Studies have shown that using excellent 1540 will produce substances that adjust self-resistance properties under unsuitable storage conditions to maintain its own quality and have good storage capacity.” please clarify. If are results of other studies, the authors have to add references, but I don’t think it is suitable for an abstract.
Response 1: We agree with the reviewer and sorry for our incorrect expression. The sentence was changed by “ The study found that under high temperature storage conditions (35 ℃), Yongyou 1540 was not significantly worse in terms of fatty acid value, whiteness value, and changes in electron microscope profile. A total of 19 key differential metabolites were screened, and lipid metabolites related to palmitoleic acid were affecting the aging of rice. At the same time, two substances, guanosine 3',5'-cyclophosphate and pipecolic acid, were beneficial to enhance the resistance of rice under harsh storage conditions, thereby delaying the deterioration of its quality and maintaining its own quality. Significant regulation of galactose metabolism, alanine, aspartate and glutamate metabolism, butyrate metabolism, and arginine and proline metabolism pathways were probably responsible for good storage capacity of Yongyou 1540. ”. Please see page 1, lines 6-15.
Point 2: The authors have performed a lot of different analysis, based on national and international protocols to analyzed the quality of the rice, along with Electron microscope profiles of rice, why the authors don’t mention them in abstract and focus only on liquid chromatography and mass spectrometry? This last technique are excellent but they aren’t the only one available and resolutive for the determination of the quality of the products. Please update the abstract and revise all the work accordingly.
Response 2: Sorry for our negligence. The measurement results of fatty acid, whiteness value, and the results observed by microscope were added in the Abstract part.
Point 3: Please increase the quality of figure 6. It not possible to read the write present in the graph.
Response 3: Sorry for our negligence. After adopting the opinions of the reviewers, the network of the remarkably perturbed metabolic pathways in Figure 6 were improved, and the resolution of the figure was adjusted by photoshop software.
Point 4: Are the authors sure that the butanoate metabolism are characteristic of the rice metabolism and not the results of microorganism-yeast metabolism of the rice and there is a contamination by these microorganisms? The change in amino acids and the increase of pipecolic acid (Pip). Moreover the authors declare: “The occurrence of Pip is common in the plant kingdom, and pathogen-induced accumulation of Pip has been reported to occur in rice…..). An analysis of microorganism contamination must be performed.
Response 4: Thanks for your reminder. Some references have demonstrated that butanoate metabolism was induced by microorganism espicially gut microbiota. For example, actinomycetes, bacteroides, clostridium, proteus, spirochetes, and thermophilic bacteria are all producers of butyric acid[1-5]. In the current study, butanoate metabolism was observed due to the fact that the content of 4-aminobutyric acid were higher in HT than CK and LT groups. 4-aminobutyric acid was a metaolite of butyric acid, which have the fuction of strengthening the brain and calming the mind, anti-epilepsy, promoting sleep, beautifying and moisturizing the skin, delaying aging and so on. In our study, 4-aminobutyric acid increased with the increase of temperature, indicating that 4-aminobutyric acid may be of great importance in improving the storage resistance as well as the nutritive value of rice.
Pipecolic acid is a common metabolites in plants, which is biosynthetic by lysine. Some studies have shown that pipecolic acid were increased during fermentation or environmental change, to produce unique taste or self protection [6-10]. In the current study, pipecolic acid were increased in the higher temperature group, which indicated that pipecolic acid may be produced in the process of high temperature fermentation, and there is another possibility, that is, the self-protection mechanism of rice against changes in the external environment.
In summary, perhaps the formation mechanism of 4-aminobutyric acid and pipecolic acid were related to microorganisms, which is a scientific idea in the following studies.
- Wang D, Guo S, He H, et al. Gut microbiome and serum metabolome analyses identify unsaturated fatty acids and butanoate metabolism induced by gut microbiota in patients with chronic spontaneous urticaria. Frontiers in cellular and infection microbiology, 2020, 10: 24.
- Xu L, Li Z, Zhuang B, et al. Enrofloxacin perturbs nitrogen transformation and assimilation in rice seedlings (Oryza sativa). Science of The Total Environment, 2022, 802: 149900.
- Zhang P, Ding Z, Zhong Z, et al. Transcriptomic analysis for Indica and Japonica rice varieties under aluminum toxicity. International journal of molecular sciences, 2019, 20(4): 997.
- Donohoe D R, Garge N, Zhang X, et al. The microbiome and butyrate regulate energy metabolism and autophagy in the mammalian colon. Cell metabolism, 2011, 13(5): 517-526.
- Yu M, Li Z, Chen W, et al. Microbiome-metabolomics analysis investigating the impacts of dietary starch types on the composition and metabolism of colonic microbiota in finishing pigs. Frontiers in microbiology, 2019, 10: 1143.
- Suharti W S, Nose A, Zheng S H. Metabolite profiling of sheath blight disease resistance in rice: in the case of positive ion mode analysis by CE/TOF-MS[J]. Plant Production Science, 2016, 19(2): 279-290.
- Sharma S, Choudhary B, Yadav S, et al. Metabolite profiling identified pipecolic acid as an important component of peanut seed resistance against Aspergillus flavus infection[J]. Journal of Hazardous Materials, 2021, 404: 124155.
- Bernsdorff F, Döring A C, Gruner K, et al. Pipecolic acid orchestrates plant systemic acquired resistance and defense priming via salicylic acid-dependent and-independent pathways[J]. The Plant Cell, 2016, 28(1): 102-129.
- Caddell D F, Louie K, Bowen B, et al. Drought shifts sorghum root metabolite and microbiome profiles and enriches the stress response factor pipecolic acid[J]. bioRxiv, 2020.
- Kang H J, Yang H J, Kim M J, et al. Metabolomic analysis of meju during fermentation by ultra performance liquid chromatography-quadrupole-time of flight mass spectrometry (UPLC-Q-TOF MS)[J]. Food Chemistry, 2011, 127(3): 1056-1064.
Point 5: The first part of the results shows a style different from the rest of the manuscript, please change.
Response 5: Thanks for your suggestion, the logic of the results part was adjusted. Two titles were set, including 2.1 Quality of rice and 2.2 Metabolomics of rice at different tempreatures, and the subheadings were set under each heading with distinct levels.
Point 6: In my opinion the conclusion section must be reported in a separate section and substantially improved to better explain the obtained results and their potentiality.
Response 6: Done. The conclusion section was added in tha main text. Please see page 11, the three part.
Reviewer 3 Report
Dear authors, I have read with interest the manuscript entitled "Effects of storage temperature on indica-japonica hybrid rice metabolites, analyzed using liquid chromatography and mass spectrometry". The work presented is interesting and some changes are required to improve it.
General comments - please remove any personal words (we, our etc) and make the text more impersonal.
The last paragraph of the Introduction need to be rewritten. You should present the aim of the research and the objectives/hypotheses. Each in a separate sentence. This will provide a step by step description of the Results and Discussion sections.
Results section - Try to make an expanded form of your results sub-sections. Only one paragraph with 5-6 lines is too short to make a sub-section. This also split too much your text.
This paragraph should be used as note under PCA graph
The abscissa PC[1] and ordinate PC[2] in the figure represent the scores of the first and second principal components, respectively. Each scatter represents a sample, and the color and shape of the scatter represent different groups. The samples were basically within the 95% confidence interval, and the QC group was tightly gathered and close to the middle of all samples, suggesting great system stability within the entire measurement queue. R2X represents the model’s interpretation of the X variable and Q2 represents the predictability of the model. The closer the two metrics are to 1, the better the model performs and the higher of the interpretability. Some parts from this paragraph are methodological considerations. Move them accordingly.
Results section should be reorganized based on above comments. In this form, the text is hard to read and understand.
Material and Methods section - present the total number of samples and any other indicators that were analyzed before the treatments. This is necessary for the replicability of the study.
What microscope did you used?
Conclusion - It should be created a separate Conclusion section, where to point your main findings with values.
Round 2
Reviewer 1 Report
I like current manuscript much more than previous submission. There are still some minor text editing errors, i.e. Latin names should be in Italic, but I believe they will be corrected after text editing.
In my opinion, manuscript may be published.
Reviewer 2 Report
The authors have answer to almost all the questions, but they do not solve the questions about the butanoate metabolism. They report only partial elements and the supplied references confirmed my opinion that is not part of rice metabolism, but of microorganism fermentation. So or authors identify the microorganisms responsible of the production of butanoate or deleted the elements related to this part from all the manuscript.
Reviewer 3 Report
Dear authors, the new version looks better and multiple points are improved. You still need to remove all the personal expressions like we, our etc.
Author Response
Response to Reviewer 3 Comments
Point 1: Dear authors, the new version looks better and multiple points are improved. You still need to remove all the personal expressions like we, our etc.
Response 1: Thanks for your suggestion. We agree with your comments and have made corrections and revisions in the text.